# Molecular mechanism of light-driven sodium pumping

Kirill Kovalev[1,2,3,4,5], Roman Astashkin[1,4], Ivan Gushchin [4], Philipp Orekhov [4], Dmytro Volkov [2,3], Egor Zinovev[2,3,4], Egor Marin [4], Maksim Rulev[2,3,6], Alexey Alekseev[2,3,4,5], Antoine Royant[1,6], Philippe Carpentier[6,7], Svetlana Vaganova[2,3], Dmitrii Zabelskii [2,3,4], Christian Baeken[2,3], Ilya Sergeev[4], Taras Balandin[2,3], Gleb Bourenkov[8], Xavier Carpena [9], Roeland Boer [9], Nina Maliar [4], Valentin Borshchevskiy[2,3,4], Georg Büldt[4], Ernst Bamberg[4,10] & Valentin Gordeliy[1,2,3,4✉]

The light-driven sodium-pumping rhodopsin KR2 from *Krokinobacter eikastus* is the only non-proton cation active transporter with demonstrated potential for optogenetics. However, the existing structural data on KR2 correspond exclusively to its ground state, and show no sodium inside the protein, which hampers the understanding of sodium-pumping mechanism. Here we present crystal structure of the O-intermediate of the physiologically relevant pentameric form of KR2 at the resolution of 2.1 Å, revealing a sodium ion near the retinal Schiff base, coordinated by N112 and D116 of the characteristic NDQ triad. We also obtained crystal structures of D116N and H30A variants, conducted metadynamics simulations and measured pumping activities of putative pathway mutants to demonstrate that sodium release likely proceeds alongside Q78 towards the structural sodium ion bound between KR2 protomers. Our findings highlight the importance of pentameric assembly for sodium pump function, and may be used for rational engineering of enhanced optogenetic tools.

[1] Institut de Biologie Structurale (IBS), Université Grenoble Alpes, CEA, CNRS, Grenoble, France. [2] Institute of Biological Information Processing (IBI-7: Structural Biochemistry), Forschungszentrum Jülich GmbH, Jülich, Germany. [3] JuStruct: Jülich Center for Structural Biology, Forschungszentrum Jülich GmbH, Jülich, Germany. [4] Research Center for Molecular Mechanisms of Aging and Age-related Diseases, Moscow Institute of Physics and Technology, Dolgoprudny, Russia. [5] Institute of Crystallography, RWTH Aachen University, Aachen, Germany. [6] European Synchrotron Radiation Facility Grenoble, Grenoble, France. [7] Institut de Recherche Interdisciplinaire de Grenoble (IRIG), Laboratoire Chimie et Biologie des Métaux (LCBM), Université Grenoble Alpes, CEA, CNRS, Grenoble, France. [8] European Molecular Biology Laboratory, Hamburg unit c/o DESY, Hamburg, Germany. [9] XALOC beamline, ALBA synchrotron (CELLS), Cerdanyola del Valles, Catalunya, Spain. [10] Max Planck Institute of Biophysics, Frankfurt am Main, Germany. ✉email: valentin.gordeliy@ibs.fr

Microbial rhodopsins (MRs) are transmembrane light-sensitive proteins, found in Archaea, Bacteria, Eukaryota, and also viruses[1]. They possess diverse biological functions and are the core of breakthrough biotechnological applications, such as optogenetics. MRs are composed of seven transmembrane α-helices (A–G) with the cofactor retinal covalently bound to the lysine residue of helix G via the Schiff base (RSB). Due to a conflict of a presence of a cation close to the RSB proton, it was believed that $Na^+$-pumping rhodopsins could not exist in nature. Despite this paradigm, the first light-driven $Na^+$ pump KR2 was identified in *Krokinobacter eikastus* in 2013[2]. Its functional and structural properties were extensively studied[3–5]. KR2 contains a characteristic for all known $Na^+$-pumping rhodopsins (NaRs) set of N112, D116, Q123 residues in the helix C (NDQ motif). It was shown that the protein pumps $Na^+$ when its concentration is much higher than that of $H^+$, which is characteristic for physiological conditions, otherwise it acts as a $H^+$ pump[6]. An extensive mutational analysis of KR2 indicated key functional residues, such as N112, D116, Q123, but also H30, S70, R109, R243, D251, S254, G263[2,3,7,8]. Moreover, potassium-pumping and potassium-channeling variants of KR2 were designed, making the protein a potential tool for optogenetics[3–5,9].

A key question remains to be answered: what is the mechanism of pumping. Indeed, principles of $Na^+$ transport by KR2 and other NaRs remain unclear. After light excitation the photocycle starts with the retinal isomerization from all-*trans* to 13-*cis* configuration[10]. In the $Na^+$-pumping mode the protein is characterized by the K, L/M, and O intermediates[10] (Fig. 1a). It is known that upon retinal isomerization the proton is translocated from the RSB to the D116 during M-state formation[2,11]. Uptake of sodium occurs in the M-to-O transition. It was hypothesized that $Na^+$ may pass the cytoplasmic gate comprised by Q123 and the neutralized RSB, and binds in the central region near D116, N112, and presumably D251 transiently in the O-state[4,10,12]. It was also suggested that with the decay of the O-state, $Na^+$ is released via R109 and the cluster of E11, E160, and R243 to the extracellular space[3,5]. Therefore, the O-state is considered to be the key for elucidation of the $Na^+$-pumping mechanism[10].

It is important that the protein always forms pentamers being reconstituted into lipid membrane[13]. KR2 pentamers appear also under physiological conditions in crystals[3,4,12] and detergent micelles[4,14]. Hence, KR2 is considered to be a pentamer in the native membrane (Fig. 1a). Pentamerization is important for $Na^+$ pumping by KR2[4]. Particularly, all previous functional investigations of KR2 were performed on the pentameric form of the protein. Moreover, it was recently reported that in the ground state under physiological conditions (PDB ID: 6REW[4]) KR2 has a large water-filled cavity near the RSB (Schiff base cavity). This conformation of the protein was called 'expanded'[4,12], and only occurs in the pentameric form of KR2. On the contrary, in the monomeric form the large cavity is absent, and the protein is in another conformation, called 'compact'[4,12]. Thus, to elucidate the mechanism of light-driven sodium pumping one needs to study the biologically relevant pentameric form of KR2, where it forms the 'expanded' conformation in the ground state.

We should note that uncovering of the mechanism of $Na^+$ pumping is of great importance and simultaneously is a challenge. First of all, it should be remarkably different from that of $H^+$ pumping[10] and, therefore, huge amount of our knowledge on the $H^+$ pumping mechanism obtained with a classic proton pump bacteriorhodopsin (BR)[15,16] could not be applied straightforward to NaRs. Not only there is a conflict of simultaneous presence of two positive charges in close proximity, the protonated RSB (RSBH$^+$) and $Na^+$ in the case of KR2, but also there are fundamental differences in the translocation of the

proton and other cations. Namely, proton transport in biological objects implies ion tunneling, which is dramatically hampered in case of larger cations. Furthermore, non-proton cation pumps cannot utilize the Grotthuss mechanism for ion translocation[15]. Hence, pathways of proton in light-driven pumps cannot be the same as those in the cation pumps like NaRs. All history of studies of BR, halorhodopsin (HR) and also sensory rhodopsin II (SRII) shows that high-resolution structures of the intermediate states of a rhodopsin are key for the understanding of the mechanisms. It is even more valid in the case of NaRs, as $Na^+$ inside the protein is absent in the ground state of KR2, which provides a wide room for speculations on the mechanism of $Na^+$ transport.

Here we present the structures of the O-state of physiologically relevant pentameric form of KR2—the key intermediate of $Na^+$ pumping—and functionally important D116N and H30A mutants of the protein. The structure of the O-state reveals $Na^+$-binding site inside the rhodopsin and together with the structures of the mutants allows us to elucidate key determinants of cation pumping by KR2.

## Results

**The structure of the O-state of pentameric KR2.** We crystallized KR2 in the functional state (at pH 8.0) using in meso approach similarly to our previous works[4,17,18]. To verify that the protein in crystals undergoes the same photocycle as in lipids, we performed time-resolved visible absorption spectroscopy on KR2 microcrystal slurries. The experiments showed that similar to the protein in detergent micelles and lipids, crystallized KR2 also forms characteristic K-, L/M- and O-states (Fig. 1). Then the O-state was trapped using an approach described in ref. [18]. In brief, we illuminated KR2 crystals with 532-nm laser to freeze-trap the O-state. The cryostream was blocked for 1 s during laser exposure and released back before turning off the laser. This procedure allows accumulation and trapping in crystals of the dominant intermediate of the protein photocycle[18,19]. Single-crystal spectrophotometry showed that after the procedure nearly all proteins in the crystal were trapped in the red-shifted intermediate state (Fig. 1b, c). KR2 undergoes only two red-shifted states during photocycle: the K- and the O-states (Fig. 1f). As the difference electron density maps (described in details below) do not indicate even a low fraction of the ground state in the structure, show the $Na^+$ bound near the RSB region, which is characteristic for the O-state of KR2, and the O-state is a dominant intermediate of the KR2 photocycle, we consider the trapped intermediate as solely the O-state.

To verify that flash-cooling does not affect the conformation of the O-state, we solved X-ray structures of the illuminated KR2 at room temperature (RT) during continuous 532-nm laser illumination using both single-crystal and serial millisecond crystallography. These approaches allow to detect the dominant conformational changes of the photocycle[19]. We collected RT crystallographic data at 2.6 Å resolution by merging of three complete datasets obtained from three single crystals (see "Methods" section and Supplementary material). We have also collected more than 1.2M diffraction patterns using the stream of mesophase containing KR2 microcrystals injected to the X-ray beam, which were processed to 2.5 Å resolution (see "Methods" section and Supplementary material). Structure refinement identified nearly 1/1 ratio of the ground/O-states populations in crystals in both cases (see Supplementary material).

The structures of the O-state at RT are identical to each other and also to that at 100 K, which indicates that cryo-cooling does not affect the O-state of KR2 (Supplementary Fig. 9). Hence, we describe further only the structure of cryo-trapped intermediate,

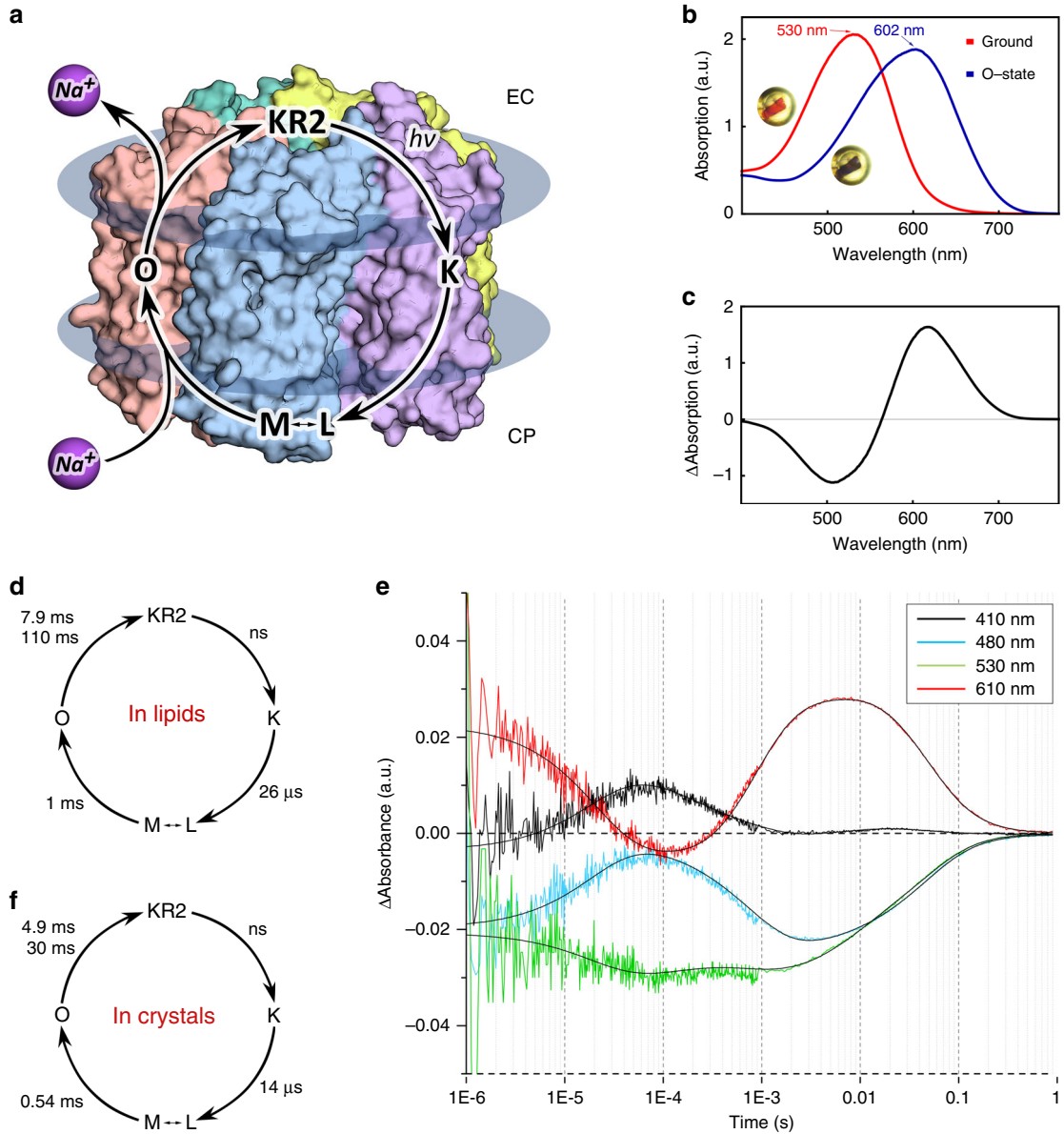

**Fig. 1 Spectroscopy of KR2 in crystals. a** Scheme of the KR2 photocycle indicates that Na[+] binding occurs transiently in the red-shifted O-state.
**b** UV–visible absorption spectra measured in crystallo at 100 K of the Ground state (red) and the O-state of KR2 (insets: photos of the same frozen KR2 crystal in the cryoloop before and after laser illumination, near the corresponding spectra). **c** Difference spectrum calculated between the blue and red spectra shown in **b**. **d** Photocycle of KR2 reconstituted in DOPC. **e** Time traces of absorption changes of KR2 microcrystals at 410 (black), 480 (light blue), 530 (green), and 610 nm (red) probe wavelengths. Black lines indicate fitting lines based on the sequential kinetic model shown in **f**. Photocycle of KR2 in microcrystals, determined in the present work.

since it has higher resolution (2.1 Å) and occupancy of the O-state (100%).

Using the crystals with the trapped intermediate, we solved the structure of the O-state at 2.1 Å (Supplementary Table 1). The crystal symmetry and lattice parameters are the same as described previously for the ground state of the protein, with one KR2 pentamer in the asymmetric unit[3,4]. The structure demonstrates notable rearrangements compared to the ground state of KR2 (Fig. 2 and Supplementary Fig. 1). The root mean square deviation (RMSD) between the backbone atoms of the pentamers and protomers of the ground (PDB ID: 6REW[4]) and the O-states (present work) are 0.55 and 0.53 Å, respectively. The main changes occur in the extracellular parts of the helices B and C, which are shifted by 1.0 and 1.8 Å, respectively (Supplementary

Fig. 1). Helices A, D and G are also displaced by 0.7 Å in the extracellular regions (Supplementary Fig. 1).

**Retinal-binding pocket of KR2 in the O-state.** The polder[20] electron density maps built around the retinal cofactor strongly suggest its all-*trans* configuration in the O-state, distorted around $C_{14}$ atom (Supplementary Figs. 2 and 3). Indeed, the fitting of the electron density maps with either 13-*cis* or the mixture of all-*trans*/13-*cis* retinal results in the appearing of strong negative peaks of the $F_o$–$F_c$ difference electron maps at the level higher than $3\sigma$. Moreover, our data shows that 13-*cis* configuration would result in the steric conflict between $C_{15}$ and $C_{20}$ atoms of the retinal and W113 and W215 residues, respectively. Therefore, all-*trans* retinal was modeled into final structure of the O-state.

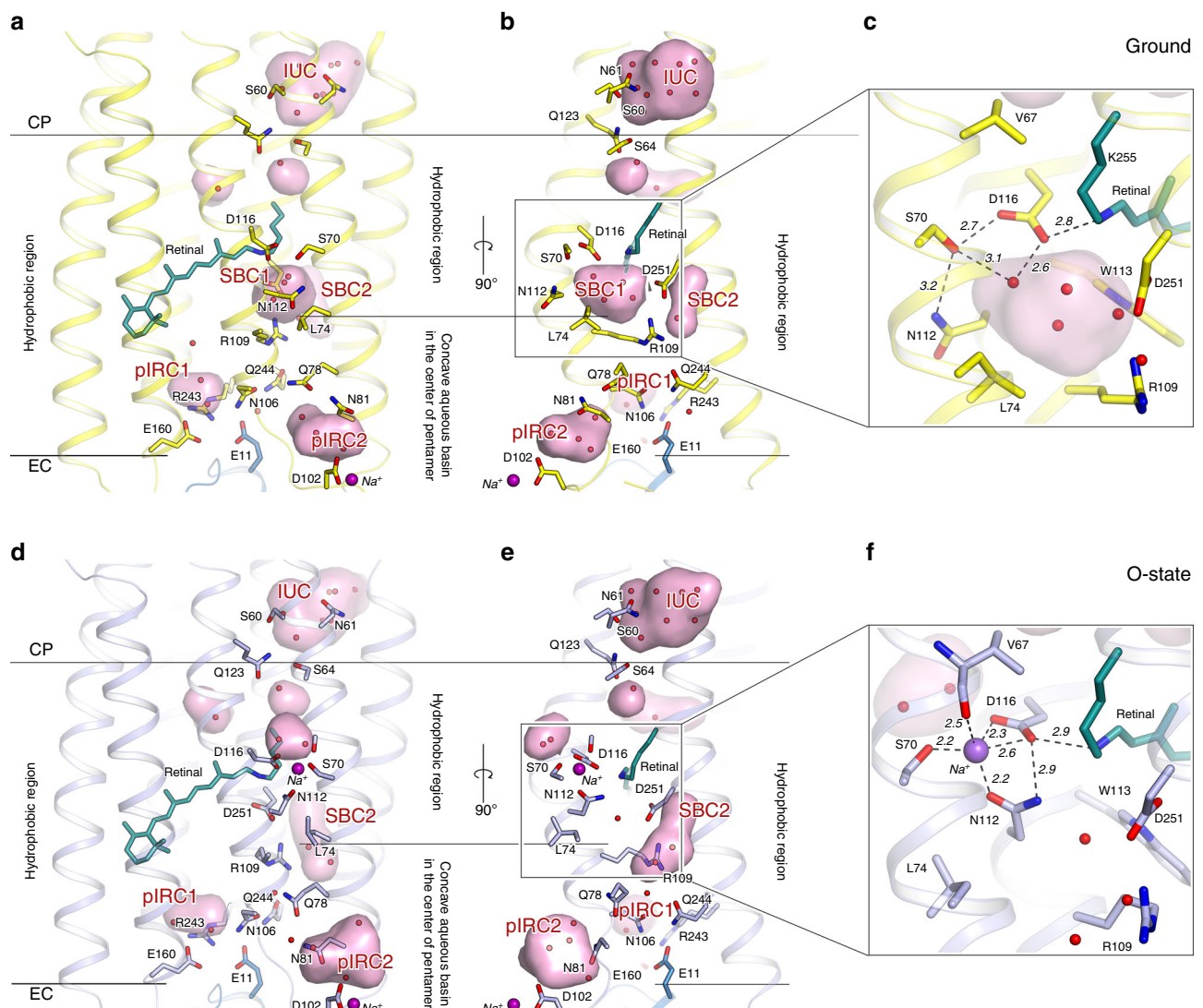

**Fig. 2 Overall comparison of the ground and O-states of KR2. a, d** Side view of the KR2 protomer in the ground (yellow, PDB ID: 6REW) and O- (blue, present work) states. **b, e** View from the side of the helices A and B. Membrane hydrophobic/hydrophilic boundaries were calculated using PPM server[56] and are shown with the black lines. The membrane boundary at the extracellular side is located at two levels for the inner and outer parts of the KR2 pentamer, respectively. Helices A and B face the concave aqueous basin, formed in the central pore of the pentamer and helices C–G face the lipid bilayer, surrounding the pentamer. Water molecules are shown as yellow and blue spheres for ground and O-state, respectively. Helices A and B are hidden for clarity. **c, f** Detailed view of the RSB region of the ground and the O-state of KR2. Cavities (ion-uptake cavity—IUC; the Schiff base cavities 1 and 2—SBC1 and SBC2, respectively; putative ion-release cavities 1 and 2— pIRC1 and pIRC2, respectively) inside the protein were calculated using HOLLOW[57] shown in pink and marked with red labels. Retinal cofactor is colored teal. Water molecules are shown with red spheres. Sodium ion is shown with a purple sphere. Hydrogen bonds involving S70, N112, D116, D251, and RSB are shown with black dashed lines. The lengths of the shown hydrogen bonds are shown with bold italic numbers and are in Å. Helix A and SBC2 are hidden for clarity.

The positions of the residues comprising the retinal pocket, particularly W113, D251, D116, I150, Y218, and W215 are correspondingly shifted relative to those in the ground state (Supplementary Fig. 3). Surprisingly, all-*trans* configuration of the retinal in the O-state is in contrast to the recently published time-resolved Fourier-transform infrared spectroscopy (FTIR) data, where authors suggested 13-*cis* configuration in the O-state of NaRs[21], but is in line with the data on another light-driven Na$^+$ pump from *Gillisia limnaea* (GLR), published in 2014, where the authors report a distorted all-*trans* configuration of retinal in the O-state[22]. In ref. [21] authors found a broad peak at 940 cm$^{-1}$ in the FTIR spectrum of the O-state of KR2 containing 12, 14-D$_2$ retinal, which was interpreted as 13-*cis* configuration of the

retinal. Our data demonstrate that although retinal is all-*trans* in both the ground and the O-state, it is kinked notably around C$_{14}$ atom only in the intermediate, but not in the ground state (Supplementary Figs. 2 and 3). This distortion may result in the appearing of the peak at 940 cm$^{-1}$ described in ref. [21]. We also cannot exclude that the inconsistency of the results on the retinal configuration in the O-state may originate from the different conditions and protein environment during the experiments.

The all-*trans* retinal configuration in the O-state of KR2 obtained in the present work means that relative location of the RSBH$^+$ and D116 side chain is similar to that in the ground state (PDB ID: 6REW[4]). RSBH$^+$ is hydrogen bonded to D116 and the distance between them is 2.9 Å in the O-state (Fig. 2f). The

existence of this hydrogen bond is supported by time-resolved resonance Raman spectroscopy[11]. Indeed, in ref. [11] authors reported that $C=N$ stretching frequencies of the RSB are very similar between the ground state $(1640\,cm^{-1})$ and the O-state $(1642\,cm^{-1})$. The $C=N$ stretching frequency is a sensitive marker for the hydrogen bond strength of the protonated RSB. The similar frequencies of the $C=N$ stretching mode support that relative locations of the $RSBH^+$ and D116 side chain are similar to those in the ground state.

**Sodium-binding site inside the protein**. The crystal structure of the O-state of KR2 clearly reveals the $Na^+$-binding site near the RSB, comprised of S70, N112, and D116 side chains and main chain oxygen of V67 (Fig. 2f and Supplementary Figs. 2 and 4). Previous mutational analysis confirms the importance of these residues for KR2 pumping activity. Indeed, D116 is crucial for KR2 functioning[2], and N112 determines ion selectivity[7]. Substitution of S70 with threonine or alanine dramatically decreases $Na^+$-pumping activity of KR2[5,9]. The mean distance between $Na^+$ and the coordinating oxygen atoms is 2.3 Å (Fig. 2f and Supplementary Fig. 2).

While in the ground state KR2 is in the 'expanded' conformation, in the O-state N112 is flipped towards S70 and D116, therefore the overall configuration is similar to that of the 'compact' conformation of KR2[4,12] (Fig. 2, Supplementary Figs. 5 and 6). This is also evidenced by disappearance of the big polar cavity near the RSB (SBC1) and enlargement and elongation of the cavity near R109–D251 pair (SBC2) in the O-state (Fig. 2 and Supplementary Fig. 6). Four water molecules, filling the SBC1 in the 'expanded' ground state (Fig. 2c), are displaced as follows: two of them are found in the small cavity formed in the intermediate near S70 at the pentamerization interface, one remains at the same place and is coordinated by N112 and D251, and the last one is moved to the SBC2 near L75 and R109 at the inner extracellular part of the protein (Fig. 2f). Upon sodium binding and formation of the 'compact' state L74 side chain also flips simultaneously with the N112 in order to avoid the steric conflict of these two residues. Our mutational analysis indicated that L74A substitution dramatically decreases pumping activity of the protein (Supplementary Fig. 7). Hence, this additionally supports the importance of the 'compact' conformation for $Na^+$ pumping by KR2.

Interestingly, the location of the sodium-binding site in the O-state of KR2 is similar to that of the chloride ion-binding sites in the ground state of light-driven chloride pumps (Supplementary Fig. 8). Namely, in a chloride-pumping rhodopsin from *Nonlabens marina* S1-08 (ClR)[23,24] the anion is coordinated by the N98 and T102 of the NTQ motif, which are analogous to the N112 and D116 of the NDQ motif of light-driven sodium pumps (Supplementary Fig. 8).

**Conformational switches guide $Na^+$ uptake and release**. The similarity of protein conformation in the O-state to the 'compact' is intriguing, however, could easily be explained. Indeed, relative location of the $RSBH^+$ and $Na^+$–$D116^-$ pair makes the distribution of the charges in the central part of the protein nearly identical to that of the KR2 with protonated D116 at acidic pH[2–4]. The structures of KR2 in the 'compact' conformation are also observed only at low pH[3–5]. It was thus suggested that the 'compact' conformation may appear in response to the D116 neutralization[4]. To understand better the nature of conformational switches in KR2 and the influence of D116 protonation on the protein conformation, we produced and crystallized KR2-D116N at pH 8.0, which mimics the WT protein with fully

protonated D116 and solved its structure in the pentameric form at 2.35 Å.

Confirming our hypothesis, the structure shows that introduction of asparagine at the position of D116 led to the flip of the side chains of N112 and L74 in comparison to the ground state of the wild type (WT) protein and disappearance of the SBC1, characteristic for the 'expanded' conformation (Supplementary Fig. 6). Overall, the structure of D116N is very similar to that of the O-state (RMSD 0.2 Å) and also to the 'compact' conformation (RMSD 0.3 Å) of the KR2-WT (Supplementary Fig. 6). However, $Na^+$ is absent inside the protomers and the relative orientation of the $RSBH^+$ and N116 is altered (Supplementary Fig. 6). Particularly, the $RSBH^+$ forms two alternative conformations, and hydrogen bond between the $RSBH^+$ and N116 is absent (see Supplementary material). We also observed that D116 protonation destabilizes the pentameric assembly of KR2 (see Supplementary material). Thus, we suggest that neutralization of D116 is the key determinant of the formation of the 'compact' conformation, which explains their structural similarity.

The structures of KR2 O-state and D116N mutant, together with previously described pH dependence of the KR2 organization[4], allow us to conclude that the 'compact' conformation and, particularly, N112 flip towards D116, stabilizes neutralized RSB counterion and correspondingly neutral transiently formed $Na^+$–$D116^-$ pair during the photocycle.

It was suggested previously that the SBC1 in the 'expanded' conformation surrounded by R109, N112, W113, D116, and D251 might be a transient $Na^+$-binding site in an intermediate state of the protein photocycle[4,12]. However, the present work shows that $Na^+$ binds far from R109 and D251 (Fig. 2f). Since the release of $Na^+$ occurs upon the O-to-ground state transition, which structurally corresponds to 'compact'-to-'expanded' switch, we suggest that the 'expanded' conformation is also important for the ion release to the extracellular space. Importantly, $Na^+$ uptake and release are guided by the switches from the 'expanded' to 'compact' and then again back to the 'expanded' conformations, respectively.

**Sodium translocation pathway**. Although the structure of the KR2 protomer near the RSB is altered in the O-state, the organization of both putative ion uptake and ion release regions remains the same to those in the ground state (Fig. 3). It is not surprising when we consider the cytoplasmic part of the protein. In the ground state $Na^+$ does not penetrate to the inside of KR2. At the same time, in the O-state $Na^+$ should not have a way to return back to the cytoplasm. Therefore, the pathway, formed upon transition from the ground to the O-state, connecting the ion uptake cavity (IUC) with the RSB region should be blocked in both ground and the O-states. Consequently, the restoration in the O-state of the initial conformation of S64–Q123 pair, separating the IUC from the RSB environment, is expected. On the other hand, the same organization of the E11–E160–R243 cluster and putative ion release cavity (pIRC1) in the ground and O-states may seem quite surprising. Previously, these residues were suggested to line the pathway of $Na^+$ release to the extracellular bulk during the direct O-to-ground transition[3–5]. The absence of the disturbance in this region means either that the energy stored in the distorted all-*trans* retinal in the O-state is enough to relocate $Na^+$ directly from the core of the protein to the bulk without any transient-binding sites or/and that there might be another ion release pathway in KR2. Indeed, the second hypothesis is supported by the mutational analysis, which showed that substitution of E11, E160, or R243 to alanines or polar non-charged residues does not abolish $Na^+$-pumping activity, however, affects the stability of the proteins[2,3,5]. The other

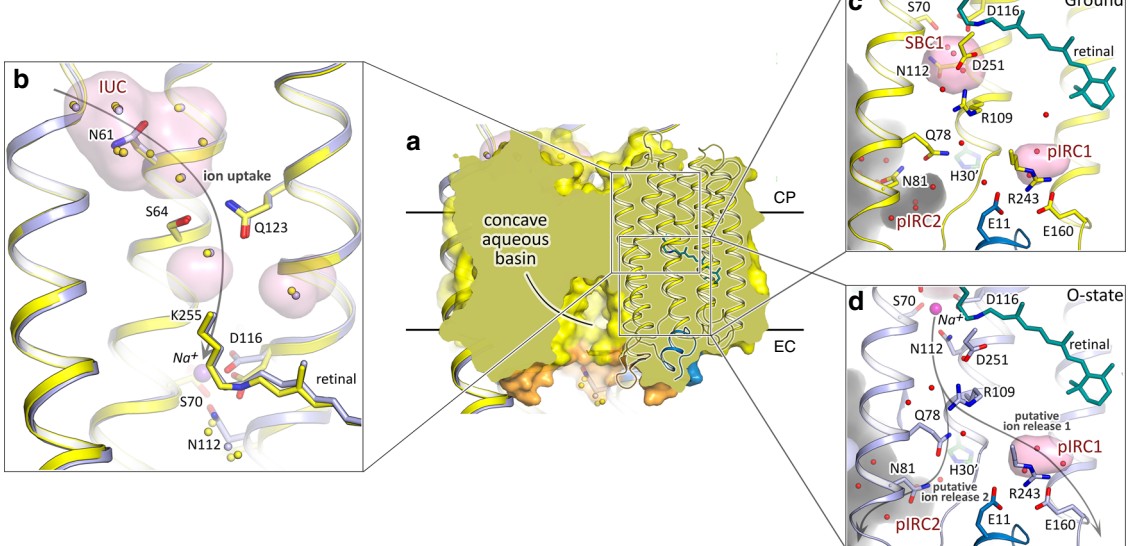

**Fig. 3 Ion uptake and release pathways of KR2. a** Section view of KR2 pentamer in the membrane. Concave aqueous basin facing the extracellular space is indicated by the black line. Only one protomer is shown in cartoon representation. Membrane core boundaries were calculated using PPM server[56] and are shown with black lines. **b** Structural alignment of the cytoplasmic parts of the ground (yellow) and O- (blue) states of KR2. Water molecules are shown with yellow and blue spheres for the ground and O-state, respectively. **c** Detailed view of the extracellular side of KR2 in the ground state. **d** Detailed view of the extracellular side of KR2 in the O-state. Cavities inside the protein are calculated using HOLLOW[57] shown in pink and marked with red labels. Protein surface concavity from the aqueous basin at the extracellular side is colored gray. Retinal cofactor is colored teal. Water molecules are shown with red spheres. Sodium ion is shown with purple sphere. N-terminal α-helix is colored blue. BC loop is colored orange. H30′ of adjacent protomer is colored with dark-green. Helices A, F, and G are hidden for clarity. Gray arrows identify putative ion uptake and two ion release pathways.

(alternative) putative way for Na$^+$ release goes from the inner extracellular part of the protein through the elongated in the O-state SBC2 to the bulk near the Na$^+$ bound at the surface of KR2 in both the ground and O-states. These channel-like pathway is constricted with the only side chain of Q78 residue (Figs. 2, 3d). The pathway propagates from the inner region between Q78, N106, and R109 to the relatively large cavity (pIRC2) between helices B and C, BC loop and helix A' of adjacent protomer at the extracellular side (Figs. 2, 3 and Supplementary Fig. 9). The cavity proceeds further to a concave aqueous basin facing the extracellular solution, formed in the central pore of the KR2 pentamer at the extracellular side and is surrounded by Q78, N81, S85, D102, Y108, and Q26' residues and filled with water molecules in both the ground and O-states (Fig. 3). Notable displacements of these residues and waters occur in the O-state, such as the flip of N81 towards H30′ of the adjacent protomer (Supplementary Fig. 10). Consequently, in the O-state additional water molecule appears in the pIRC2, which is coordinated by hydrogen bonds with E26′, H30′, and N81 (Supplementary Fig. 10). The positions of Q78 and Y108 are also altered in the O-state (Supplementary Fig. 10).

To probe the possible ion release pathways, we conducted 10 short metadynamics simulations starting from the sodium-bound conformation (Supplementary Fig. 11 and Supplementary Movie 1). The simulations revealed that the R109 sidechain forms a barrier for sodium exiting SCB1, and changes its position to allow sodium passage. Upon passing R109, Na$^+$ either exits the protein via pIRC2 (8 simulations out of 10), or proceeds towards pIRC1 (two simulations). In the first scenario, the ion is quickly released towards the aqueous basin in the middle of KR2 pentamer in the vicinity of another ion found at the interface between the protomers, and is sometimes observed to replace it (Supplementary Movie 1). In the second scenario, the ion samples different locations around the E11–E160–R243 triad and is later released via N106 and Q157 on the outer side of the pentamer.

Importantly, the organization of the pIRC2 region is identical in the O-state and KR2-D116N (Supplementary Fig. 12A). It means that the rearrangements on the surface of the KR2 occur not directly in response to the retinal isomerization upon photon absorption, but rather due to the redistribution of charges in the central inner part of the protein protomer. Such long-distance interactions between the RSB-counterion and pentamer surface were already studied for the WT and H30A variant of KR2[14,25]. Recently, it was also shown that there is an allosteric communication between the interprotomer Na$^+$-binding site and the RSB hydrogen bond already in the ground state[26]. Thus, structural rearrangements of the RSB-counterion pair upon Na$^+$ release and corresponding 'compact'-to-'expanded' conformational switch may allosterically affect the interprotomer Na$^+$-binding site, promoting Na$^+$ unbinding from the site, observed in metadynamics simulations.

To gather more details about long-distance interactions between the protein core and surface, we solved the structure of pentameric form of KR2-H30A at pH 8.0 at 2.2 Å.

Overall, the structure of this mutant is nearly the same as that of the ground state of WT protein (RMSD 0.15 Å) (Supplementary Figs. 5, 13). The organization of their inner cytoplasmic, central, and extracellular parts is identical. Surprisingly, in contradiction to the earlier FTIR experiments, Na$^+$ bound at the oligomerization interface of the WT protein is also present in H30A[2]. However, the region of H30 is altered (Supplementary Fig. 12). Particularly, H30 side chain is replaced by two additional water molecules (w$_B$′ and w$_B$″) (Supplementary Fig. 12). Moreover, the H30A mutation leads to appearance of second alternative conformation of Y108, not identified in other structures of KR2 or its variants (Supplementary Fig. 12). It was shown that H30A is more selective to sodium and almost does not pump protons[2]. As the only existing differences in the structures of the WT and the mutant occur near Q78, N81, Y108, H30′ and pIRC2, we suggest that this region is important for

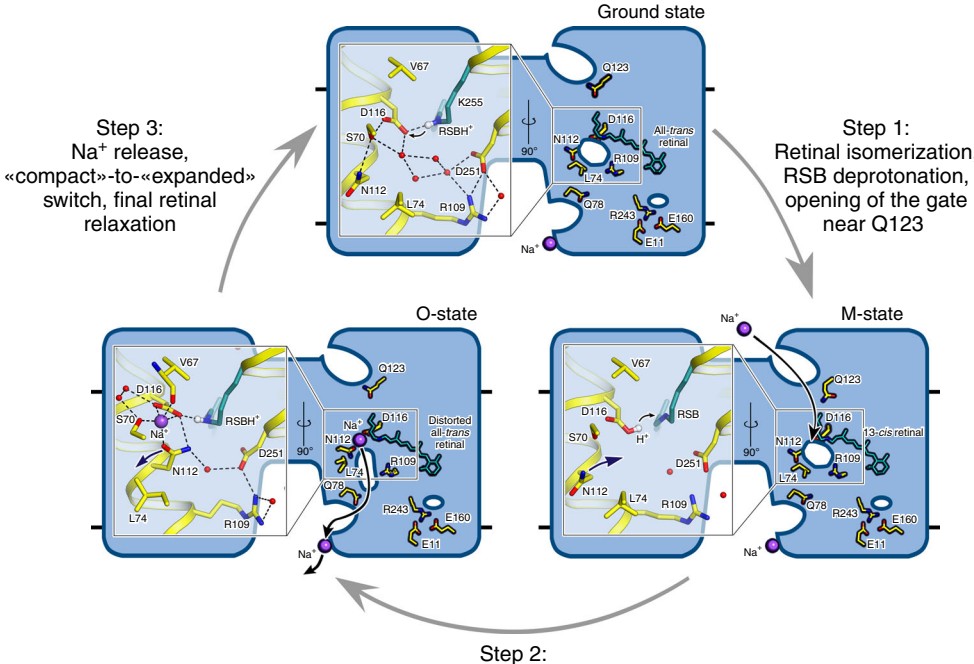

**Fig. 4 Proposed Na⁺ pumping mechanism.** Schematic side section view of the KR2 pentamer is shown. Membrane core boundaries are shown with black lines. Cavities are demonstrated as white ellipses. Enlarged view of the RSB region is shown at the left part of the pentamer. The 13-*cis* configuration of the retinal cofactor is modeled manually for schematic representation. Na⁺ is shown with violet spheres. Black arrows indicate proposed Na⁺ uptake and release pathways. Violet arrows indicate rearrangement of N112 side chain during "expanded"-to-"compact" and back "compact"-to-"expanded" switches. Small gray arrows indicate the translocation of the hydrogen from the Schiff base to D116 during the formation of the M-state and following reprotonation of the Schiff base from the D116 in the M-to-O transition. Retinal cofactor is colored teal. Waters are shown with red spheres. Hydrogen bonds in the RSB region are shown with black dashed lines.

cation selectivity, which additionally supports the hypothesis that this region is a part of ion translocation pathway.

To verify the suggested ion-release pathway, we performed functional studies of KR2 mutants in *E. coli* cells suspension, similar to previous works[3,27] (Supplementary Fig. 7). The results showed that Q78 is the key residue, which is likely to act as a gate for sodium, flipping upon sodium passage (Q78L mutant remains almost fully functional). The blocking of Q78 motion (Q78Y,W mutations) resulted in dramatic decrease of the pumping activity. In Q78A mutant, similar to Q123A, the sodium pumping activity is retained, however decreased notably in comparison to the WT protein. Another interesting finding was that Y108A mutant almost fully lost its pumping ability.

Hence, we suggest that Na⁺ translocation pathway propagates from IUC to pIRC2 via a chain of polar inner cavities, which are modified during photocycle. IUC and pIRC2 are separated from the inside of KR2 by two weak gates near Q123 and Q78, respectively. This makes the KR2 ion pathway similar to that of the channelrhodopsin 2[28] (*Cr*ChR2) (Supplementary Fig. 14). However, unlike in *Cr*ChR2, KR2 has the Na⁺-binding site in the central region near the RSB.

**Mechanism of sodium pumping.** Crystal structures of the O-state and D116N and H30A mutants of KR2 together with available literature data allow us to suggest a mechanism of protein functioning (Fig. 4).

Step 1: In the ground state KR2 is in the 'expanded' conformation with the SBC1 filled with four water molecules. The RSBH⁺ is hydrogen bonded to its counterion D116⁻. The Q123-S64 gate separates the IUC and the RSB. With the

absorption of the light photon the retinal isomerizes from all-*trans* to 13-*cis* configuration and the red-shifted K-state appears in nanoseconds, followed by the formation of the L/M intermediate in about 30 μs. The proton is translocated from the RSBH⁺ to the D116⁻ with the formation of the M-state and the hydrogen bond between them is absent in this intermediate. The Q123-S64 gate also opens in this step.

Step 2: With the rise of the O-state Na⁺ passes the gate and deprotonated RSB, and binds between S70, N112, and D116 belonging to helices B and C. Na⁺ uptake results in the proton translocation from D116 back to the RSB with the restoration of the RSB-counterion hydrogen bond, thus preventing the Na⁺ backflow to the cytoplasmic side. It also causes the flip of N112 side chain for the stabilization of Na⁺–D116⁻ pair, therefore, the 'compact' state appear at this step. Retinal is still in the distorted all-*trans* conformation in the O-state.

Step 3: With the decay of the O-state retinal returns completely to its ground configuration and the 'expanded' conformation of KR2 occurs with the N112 flip back to the pentamerization interface, thus opening the way for Na⁺ release. In the O-state the release pathway is prepared as it is evidenced from the O-state structure. We suggest that the release preferentially involves Q78 and proceeds towards the cavity, formed by N81, Y108, and H30′ of adjacent protomer (pIRC2). We also suggest that four water molecules, filling the SBC1 in the dark state are involved in the Na⁺ hydration during its transitions inside the protein. The Na⁺ release might proceed using the relay mechanism, when the ion, released from the KR2 protomer, replaces the ion, bound at the protein oligomerization interface. The proposed relay mechanism allows lowering the energy barriers for facilitation of the Na⁺

release. The E11–E160–R243 cluster and the cavity near it (pIRC1), suggested earlier to be involved in Na$^+$ release, thus may be involved mainly in protein stabilization, rather than in ion translocation pathway.

Last but not least, in the absence of Na$^+$ KR2 acts as a proton pump with the significantly altered photocycle and the absence of the pronounced O-state. The long living L/M/O-like state decays slowly in the proton-pumping mode[2]. We note that this is in agreement with the suggested mechanism of Na$^+$ pumping. We suggest that the formation of the K-, L- and M-states is the same for the Na$^+$- and H$^+$-pumping modes, however, in the absence of Na$^+$ the ion does not flow diffusely to the central region when the RSB is neutral, and RSB is presumably reprotonated from the cytoplasmic side through the IUC and no rearrangements occur in the region of D116. Then the slow relaxation of the retinal to all-*trans* configuration triggers the proton release from the D116 to the extracellular side and the protein returns to the ground state.

## Discussion

The presented here structure of the key functional O intermediate state of the KR2 rhodopsin allowed us to propose the pathway and molecular mechanism of the active light-driven Na$^+$ transport. We suggest that the pathway of Na$^+$ connects two aqueous concavities in the cytoplasmic and extracellular parts of the KR2, respectively, via several polar cavities inside the protein, which are separated from each other by weak gates. From that side, the organization is very similar to that of *Cr*ChR2[28]. The principal difference is that in opposite to *Cr*ChR2, KR2 has the tight transient Na$^+$-binding site near the Schiff base, separating the cytoplasmic and extracellular parts of the protein. Judging from the arrangement of the cavities in the O-state, Na$^+$ transport in KR2 appears rather like in a perfectly outwardly directed Na$^+$ channel than in a pump, which we interpret as a chimera between the two fundamental mechanisms. Similar concept was suggested earlier[10], however, it mostly concerned the diffusion Na$^+$ uptake from the cytoplasmic part. The presence of the positively charged R109 side chain at the extracellular part of the protein presumably prevents Na$^+$ uptake from that side in the M-state, as also evidenced by a recent finding that R109Q mutation converts KR2 into a potassium channel[9]. Presented here model of the O-state together with metadynamics simulations and mutational analysis, as well as the absence of any intermediate states during O-to-ground transition in KR2, strongly suggest that Na$^+$ release proceeds also on the path of least resistance without any notable structural rearrangements of the protein. This is supported by the fact that the retinal in the O-state is almost completely relaxed to its initial configuration. We also speculate that for the facilitation of the Na$^+$ release (lowering of the energy barrier for the Na$^+$ release against electrochemical gradient), the protein use the relay mechanism, in which the released ion replaces the ion bound at the oligomerization interface. The fact was also evidenced by the metadynamics simulations, described in the present manuscript and by the recently reported existence of the allosteric communications between the RSB and interprotomer ion-binding site[26]. Current understanding of the Na$^+$ release from the KR2 is summarized in the Supplementary Movie 1.

## Methods

**Protein expression and purification.** *E. coli* cells of strain SE1 (Staby™Codon T7, Eurogentec, Belgium) were transformed with the KR2 expression plasmid. Transformed cells were grown at 37 °C in shaking baffled flasks in an auto-inducing medium ZYP-5052[29] containing 100 mg/L ampicillin. When glucose level in the growing bacterial culture dropped below 10 mg/L, 10 µM all-*trans*-retinal (Sigma-Aldrich, Germany) was added, the incubation temperature was reduced to 20 °C and incubation continued overnight. Collected cells were disrupted in M-110P Lab Homogenizer (Microfluidics, USA) at 25,000 psi in a buffer containing

20 mM Tris–HCl pH 8.0, 5% glycerol, 0.5% Triton X-100 (Sigma-Aldrich, USA) and 50 mg/L DNase I (Sigma-Aldrich, USA). The membrane fraction of cell lysate was isolated by ultracentrifugation at $90,000 \times g$ for 1 h at 4 °C. The pellet was resuspended in a buffer containing 50 mM NaH$_2$PO$_4$/Na$_2$HPO$_4$ pH 8.0, 0.1 M NaCl and 1% DDM (Anatrace, Affymetrix, USA) and stirred overnight for solubilization. Insoluble fraction was removed by ultracentrifugation at $90,000 \times g$ for 1 h at 4 °C. The supernatant was loaded on Ni-NTA column (Qiagen, Germany) and KR2 was eluted in a buffer containing 50 mM NaH$_2$PO$_4$/Na$_2$HPO$_4$ pH 7.5, 0.1 M NaCl, 0.5 M imidazole and 0.1% DDM. The eluate was subjected to size-exclusion chromatography on 125 ml Superdex 200 PG column (GE Healthcare Life Sciences, USA) in a buffer containing 50 mM NaH$_2$PO$_4$/Na$_2$HPO$_4$ pH 7.5, 0.1 M NaCl, 0.05% DDM. Protein-containing fractions with the minimal $A_{280}/A_{525}$ absorbance ratio were pooled and concentrated to 60 mg/ml for crystallization.

**Measurements of pumping activity in E. coli cells.** *E. coli* cells of strain C41 (DE3) (Lucigen, USA) were transformed with the KR2 expression plasmid. Transformed cells were grown at 37 °C in shaking baffled flasks in an autoinducing medium, ZYP-5052[29] containing 100 mg/L ampicillin, and were induced at optical density OD600 of 0.7–0.9 with 1 mM isopropyl β-D-1-thiogalactopyranoside (IPTG) and 10 µM all-*trans*-retinal. 3 h after induction, the cells were collected by centrifugation at $4000 \times g$ for 10 min and were washed three times with unbuffered salt solution (100 mM NaCl, and 10 mM MgCl$_2$) with 30-min intervals between the washes to allow exchange of the ions inside the cells with the bulk. After that, the cells were resuspended in 100 mM NaCl solution and adjusted to an OD$_{600}$ of 8.0. The measurements were performed on 3 ml of stirred cell suspension kept at 1 °C. The cells were illuminated for 5 min with a halogen lamp (Intralux 5000-1, VOLPI) and the light-induced pH changes were monitored with a pH meter (LAB 850, Schott Instruments). Measurements were performed with the addition of 30 µM of protonophore carbonyl cyanide 3-chlorophenylhydrazone (CCCP).

**Crystallization details and crystals preparation.** The crystals were grown using the in meso approach[30,31], similarly to our previous work[27,32,33]. The solubilized protein in the crystallization buffer was added to the monooleoyl-formed lipidic phase (Nu-Chek Prep, USA). The best crystals were obtained using the protein concentration of 25 mg/ml. The crystals of monomeric (for D116N mutant) and pentameric (for WT protein and H30A mutant) forms were grown using the precipitate 1.0 M sodium malonate pH 4.6 and 1.2 M sodium malonate pH 8.0, respectively (Hampton Research, USA). Crystallization probes were set up using the NT8 robotic system (Formulatrix, USA). The crystals were grown at 22 °C and appeared in 2–4 weeks. Before harvesting, crystallization drop was opened and covered with 3.4 M sodium malonate solution, pH 8.0, to avoid dehydration. All crystals were harvested using micromounts (MiTeGen, USA) and were flash-cooled and stored in liquid nitrogen for further crystallographic analysis.

**Time-resolved visible absorption spectroscopy.** The laser flash photolysis was performed similar to that described by Chizhov et al.[34,35] with minor differences. The excitation system consisted of Nd:YAG laser Q-smart 450mJ with OPO Rainbow 420–680 nm range (Quantel, France). For the experiments the wavelength of the laser was set 525 nm. Microcrystals of KR2 in the lipidic cubic phase were plastered on the $4 \times 7$ mm cover glass. The thickness of the slurries was adjusted in order to give sufficient signal. The glass with crystal slurries was placed into $5 \times 5$ mm quartz cuvette (Starna Scientific, China) filled with the buffer solution containing 3.4 M sodium malonate pH 8.0 and thermostabilized via sample holder qpod2e (Quantum Northwest, USA) and Huber Ministat 125 (Huber Kältemaschinenbau AG, Germany). The detection system beam emitted by 150 W Xenon lamp (Hamamatsu, Japan) housed in LSH102 universal housing (LOT Quantum Design, Germany) passed through pair of Czerny–Turner monochromators MSH150 (LOT Quantum Design). The received monochromatic light was detected with PMT R12829 (Hamamatsu). The data recording subsystem represented by a pair of DSOX4022A oscilloscopes (Keysight, USA). The signal offset signal was measured by one of oscilloscopes and the PMT voltage adjusted by Agilent U2351A DAQ (Keysight). The time-resolved visible absorption spectroscopy data was processed and fitted using OriginPro8.5 software.

**Accumulation of the intermediate state in KR2 crystals.** Absorption spectra of KR2 in solution were collected using the UV-2401PC spectrometer (Shimadzu, Japan) using the UVProbe 2.33 software. The spectroscopic characterization of O-state build-up in KR2 crystals was performed at the icOS Lab located at the ESRF[36]. The same set up was established at the P14 beamline of the PETRAIII synchrotron source (Hamburg, Germany) for accumulation of the O-state in crystals for X-ray diffraction data collection. Also the same accumulation procedure was applied to crystals at icOS and P14 beamline. Briefly, UV–visible absorption spectra were measured using as a reference light that of a DH-200-BAL deuterium-halogen lamp (Ocean Optics, Dunedin, FL) connected to the incoming objective via a 200 µm diameter fiber, resulting in a 50 µm focal spot on the sample, and a QE65 Pro spectrometer (Ocean Optics, Dunedin, FL) connected to the outgoing objective via a 400 µm diameter fiber. The actinic light comes from a 532 nm laser (CNI Laser, Changchun, P.R. China) coupled to a 1000 µm diameter fiber which is connected to the third objective whose optical axis is perpendicular to those of the ingoing and

outgoing objectives. Ground states spectra (100 ms acquisition time averaged 20×) were collected on crystals flash-cooled in liquid nitrogen and kept under a cold nitrogen stream at 100 K. In order to maximize the population of the O-state, a crystal was put under constant laser illumination at 100 K, the nitrogen stream was then blocked for 2 s, then the laser was switched off once the crystal is back at 100 K. For the accumulation of the O-state laser power density of 7.5 mW/cm$^2$ at the position of the sample was used. The mean size of the crystals was $200 \times 100 \times 30\ \mu m^3$ (Supplementary Fig. 22). The plate-like crystals were oriented so that the largest plane ($200 \times 100\ \mu m^2$) was as perpendicular to the laser beam. The laser beam was focused to the size of $500 \times 500\ \mu m^2$ ($1/e^2$). A UV–visible absorption spectrum was then recorded to show the red-shifted absorption maximum characteristic of the O-state of KR2. The crystals with accumulated intermediate state were then stored in liquid nitrogen and transported to the PETRAIII, Hamburg, Germany for the X-ray experiments and showed the same structure as that obtained using crystals with the O-state, accumulated directly at the P14 beamline of PETRAIII.

**Acquisition and treatment of diffraction data.** X-ray diffraction data of D116N and H30A mutants were collected at the beamlines ID23-1 and ID29 of the ESRF, Grenoble, France, using a PILATUS 6M and EIGER 16M detectors, respectively. The data collection was performed using MxCube2 software. X-ray diffraction data of the KR2 O-state at 100 and 293 K (room temperature, RT) was collected at the P14 beamline of the PETRAIII, Hamburg, Germany, using EIGER 16M detector. For the collection of the X-ray diffraction data at RT the crystals in the cryoloop were placed on the goniometer of P14 beamline and maintained in the stream of the humid air (85% humidity). The stream of humid air was provided by the HC humidity controller (ARINAX, France). For activation of the proteins in crystals and obtaining the structure of the O-state at RT the laser flash was synchronized with the X-ray detector. The laser was illuminating the crystal only during X-ray data collection to avoid drying and bleaching. The crystals were rotated during the data collection and laser illumination. Diffraction images were processed using XDS[37]. The reflection intensities of the monomeric form of D116N mutant were scaled using the AIMLESS software from the CCP4 program suite[38]. The reflection intensities of all the pentameric forms were scaled using the Staraniso server[39]. There is no possibility of twinning for the crystals. For the both structures of KR2 at RT diffraction data from three crystals was used (Supplementary Table 1). In all other cases, diffraction data from one crystal was used. The data statistics are presented in Table S1.

**Serial crystallography data collection and processing.** Serial milisecond crystallography data of the O-state of KR2 was obtained at RT at BL13-XALOC beamline of ALBA (Barcelona, Catalunya) using a PILATUS 6M detector working at 12.5 Hz and a $60 \times 40\ \mu m$ FWHM (HxV) sized beam at 12.6 keV ($1.4 \times 10^{12}$ ph/s). For that purpose, an LCP stream of protein microcrystals was injected into the focus region using a LCP injector[40], placed at 45° of the diffractometer table, with the help of an ÄKTA pump flowing at 1 μl/min and a constant helium supply (10–14 psi) yielding an extrusion speed of 30 nl/min for 100 μm capillary. A Roithner laser source (RLTMLL-532-100-5) working at 20 mW was used for protein activation.

A total number of 1,208,640 detector images were collected and processed with CrystFEL (version 0.8.0)[41] without any additional modification. Among all images collected, 350,862 were identified as potential crystal hits with more than 30 Bragg peaks with min-snr = 3.5, threshold = 12, highres = 2.5 parameters using peakfinder8 algorithm as implemented in CrystFEL, corresponding to an average hit rate of 29%. The overall time of data collection from a sample with a total volume of 120 μl was about 36 h.

Data were processed using CrystFEL. For peak finding, we used peakfinder8 and min-snr = 3.5, threshold = 12. For indexing, indexers dirax[42], xds[37], taketwo[43] (in that order) were used, with --multi option enabled. Integration was performed using int-radius = 3,4,5. Data were merged using process_hkl with min-res = 3.3, push-res = 3.0, and symmetry = mmm. This yielded a dataset with 131,872 indexed images with 136,656 crystals, corresponding to 39% average indexing rate. Among these images, 38,761 were merged together after process_hkl rejection. Among indexers, dirax was the most successful one, providing 77,681 indexed crystals. Initial geometry, provided by the beamline staff, was optimized with detector-shift and geoptimiser, as described in ref. [44].

**Structure determination and refinement.** Initial phases for the pentameric structures were successfully obtained in the C222$_1$ space group by molecular replacement (MR) using MOLREP[45] using the 6REW structure as a search model. Initial phases for monomeric KR2-D116N were successfully obtained in the I222 space group by MR using the 4XTL structure as a search model. The initial MR models were iteratively refined using REFMAC5[46], PHENIX[47], and Coot[48].

**Molecular dynamics simulations.** The simulation system consisted of a KR2 pentamer in the O-state with sodium ions in SBC2 and cocrystallized water molecules. The proteins was embedded in a POPC bilayer (256 lipids) and then solvated with TIP3P water with a Na+/Cl− concentration of 150 mM using the CHARMM-GUI web-service[49]. The simulation box contained 94,817 atoms in total. All ionizable amino acids were modeled in their standard ionization state at pH 8, including D116 and D251 which were modeled charged.

The CHARMM-GUI recommended protocols were followed for the initial energy minimization and equilibration of the system. The atoms of protein and lipids in the system were subjected to a harmonic positional restraint and 5000 steps of steepest descent minimization followed by two 25 ps equilibration steps in the NVT ensemble using Berendsen thermostat and one 25 ps and three 50 ps equilibration steps in the NPT ensemble using Berendsen thermostat and barostat. During all equilibration steps, the force constants of the harmonic positional restraints were gradually reduced to zero. The system was further equilibrated for 10 ns in the NPT ensemble with Nose–Hoover thermostat and Parrinello−Rahman barostat, which were also used for the further production simulations. The temperature and pressure were set to 303.3 K and 1 bar with temperature and pressure coupling time constants $\tau t = 1.0\ ps^{-1}$ and $\tau p = 0.5\ ps^{-1}$, respectively. All MD simulations were performed with GROMACS version 2018.1[50]. The time step of 2 fs was used for all the simulations except for the early steps of equilibration. The CHARMM36 force field[51] was used for the protein, lipids, and ions. Parameters for retinal bound to lysine were adapted from ref. [52].

In order to investigate the putative sodium translocation pathways, metadynamics approach (metaMD) was employed[53]. This method is based on biasing of the potential surface via addition of repulsive functions ("hills", typically Gaussians) which force the investigated molecular system to explore its configurational space broader and faster than in a regular unbiased MD simulation. Since the correct sampling of protein degrees of freedom relevant to the ion release at the late states of the photocycle appears problematic, the performed metadynamics simulations allowed us to reveal possible sodium unbinding pathways rather than quantitatively assess the free energies changes associated with them. We used the PLUMED plugin for GROMACS to perform metaMD simulations[54]. The projections of the vector connecting the sodium ion and its original position onto x, y, and z directions were used as the collective variables (i.e., 3 CVs were used). In order to prevent sodium passage back to the cytoplasmic side we applied a flat-bottom potential ($k = 1000$ kJ/mol/nm$^2$) in the normal to the membrane direction, which discouraged ion moving towards the cytoplasmic side. Also, harmonic restrains were applied to all protein C$\alpha$ atoms above the C$\alpha$ atom of K255 (in the direction of CP) to prevent the overall motion of the protein complex. The deposition rate for hills was 0.5 ps; the width and height of deposited hills were equal to 0.05 nm and 1 kJ/mol, respectively. The simulations were continued until the exit of ion from the protein interior was observed (typically, during 5–20 ns). We have carried out 10 metaMD runs in total, 2 replicates for each of the 5 protomers of the KR2 pentamer.

**Reporting summary.** Further information on research design is available in the Nature Research Reporting Summary linked to this article.

## Data availability

Data supporting the findings of this manuscript are available from the corresponding author upon reasonable request. A reporting summary for this Article is available as a Supplementary Information file. The source data underlying Fig. 1b, c, e and Supplementary Figs. 7 and 21 are provided as a Source Data file. Crystallographic data that support the findings of this study have been deposited in the Protein Data Bank (PDB) with the accession codes: PDB: 6XYT (the O-state of KR2 at 100K), PDB: 6YBY (D116N mutant in monomeric form), PDB: 6YBZ (D116N mutant in pentameric form), PDB: 6YC0 (steady-state-SMX activated state of KR2 at RT), PDB: 6YC1 (H30A mutant in pentameric form), PDB: 6YC2 (dark-state of KR2 at RT), PDB: 6YC3 (dark-state of KR2 at 100K) PDB: 6YC4 (steady-state activated state of KR2 at RT). Serial crystallography data corresponding to the 6YC0 model have been deposited to the Coherent X-ray Imaging Data Bank[55] with CXIDB ID 141. These include detector.cbf files without any additional treatment for images with more than 30 diffraction peaks, as described in "Methods" section.

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

## Acknowledgements

We acknowledge the Structural Biology Group of the European Synchrotron Radiation Facility (ESRF) and The European Molecular Biology Laboratory (EMBL) unit in Hamburg at Deutsche Elektronen-Synchrotron (DESY) for granting access to the synchrotron beamlines. Special thanks are given to BL13-XALOC technician, ALBA floor coordinators and, most specially, to J.M. Martín-García (Centre for Applied Structural Discovery-ASU, US), for their contribution to adapt the injector to XALOC Beamline specifications. This work was supported by the common program of Agence Nationale de la Recherche (ANR), France and Deutsche Forschungsgemeinschaft, Germany (ANR-15-CE11-0029-02), Ministry of Science and Higher Education of the Russian Federation (075-00337-20-03/FSMG-2020-0003) and by funding from Frankfurt: Cluster of Excellence Frankfurt Macromolecular Complexes (to E.B.) by the Max Planck Society (to E.B.) and by the Commissariat à l'Energie Atomique et aux Energies Alternatives (Institut de Biologie Structurale)–Helmholtz- Gemeinschaft Deutscher Forschungszentren (Forschungszentrum Jülich) Special Terms and Conditions 5.1 specific agreement. This work used the icOS and HTX platforms of the Grenoble Instruct-ERIC center (ISBG; UMS 3518 CNRS-CEA-UJF-EMBL) within the Grenoble Partnership for Structural Biology (PSB). Platform access was supported by FRISBI (ANR-10-INBS-05-02) and GRAL, a project of the University Grenoble Alpes graduate school (Ecoles Universitaires de

Recherche) CBH-EUR-GS (ANR-17-EURE-0003). Data collection, data treatment and structure solution were supported by RFBR (19-29-12022).

## Author contributions

C.B., S.V., and D.Z. expressed and purified the proteins; T.B. supervised the expression and purification; R.A. and K.K. crystallized the proteins; D.Z. helped with crystallization; K.K. collected absorption spectra from crystals and performed cryo-trapping of the intermediate; A.R. and P.C. supervised the absorption spectra collection; D.V. performed flash photolysis experiments on KR2 crystals and processed the data; K.K. helped with the flash photolysis experiments; K.K. collected the diffraction data with the help of R.A., D.Z., and solved the structures; X.C. and R.B. performed serial crystallography data acquisition at XALOC; E.Z. and E.M. processed the serial crystallography data; I.G. supervised structure refinement and analysis; G.B. helped with data collection; P.O. performed the molecular dynamics simulations; I.S. prepared the systems for simulations; M.R. performed oligomerization analysis of the proteins; A.A., N.M., V.B., E.B., and G.B. helped with data analysis, V.G. supervised the project; K.K. and V.G. analyzed the results and prepared the manuscript with input from all the other authors.

## Competing interests

The authors declare no competing interests.
