## [Peer Review File · Nature Communications]

Peer Review File – first round comments:

Reviewers' comments:

Reviewer #3 (Remarks to the Author):

The authors have incorporated all changes requested by me. I only have two minor changes to add before the manuscript is ready for publication concerning the Computational part:

-Methods: please give height and width of Gaussians that were placed in metadynamics runs.

-line 253: change "molecular dynamics" to "metadynamics". Furthermore include a sentence that you performed these simulations with the intent to find unbinding pathways, but not to determine free energies along them.

Steffen Wolf, University of Freiburg, Germany.

Reviewer #4 (Remarks to the Author):

In this paper, the authors present the structures of the O-state of the light-driven sodium ion-pumping rhodopsin KR2. The O-state is the key intermediate of Na⁺ pumping in KR2. The authors showed the protein structure for a physiologically relevant pentameric form. They identified the transient Na⁺ binding site inside the rhodopsin in the O-state and structural differences between the ground and the O-states. Together with the structures of the mutants, the authors discussed the protein structures to elucidate key determinants of cation pumping by KR2. The structural data provided in this paper are highly important and well-analyzed to understand the pathway of Na⁺ and the ion-pumping mechanism of KR2. Insights obtained by recent spectroscopic studies are closely related to the pathway and the ion-pumping mechanism proposed in the present study. Although the discussion in the manuscript is sound, the manuscript will be improved by the addition of the relating insights provided the spectroscopic studies. I suggest that I recommend the publication of this work in Nature Communications after major revisions indicated as follows.

1. Line 145. The authors claimed that the retinal chromophore adopts a distorted all-trans configuration in the O-state. However, this is in contrast to the recently published time-resolved FTIR data on KR2. The discrepancy should be discussed more. Is the probability that the electron density is attributed to the 13-cis form completely ruled out? The authors mentioned the chromophore configuration in the O-state of GLR. But, this is a different protein. The discrepancy between the data reported for KR2 should be discussed more clearly.

2. Line 152. The authors claimed that relative locations of the RSBH⁺ and D116 side chain are similar to those in the ground state. This conclusion is quantitatively supported by the data of a recent resonance Raman study (Ref. #14). Ref. #14 reported that C=N stretching frequencies of the RSB are very similar between the ground state (1640 cm⁻¹) and the O-state (1642 cm⁻¹). The C=N stretching frequency is a sensitive marker for the hydrogen bond strength of the protonated RSB. The similar frequencies of the C=N stretching mode support the authors' claim that relative locations of the RSBH⁺ and D116 side chain are similar to those in the ground state. I suggest to include the above-mentioned discussion on the C=N stretching frequencies in the manuscript.

3. Line 184. The switching between the expanded and the compact conformations is interesting and reasonable to understand the ion pumping mechanism. A recent resonance Raman study (Otomo et al., *Biochemistry* 2020, 59, 520–529) showed that the Na⁺ binding to the interprotomer binding site changes the hydrogen bond strength of the protonated RSB in the ground state of KR2, suggesting an allosteric communication from the interprotomer binding site to the protonated RSB. If the interprotomer binding site and the RSB are allosterically coupled, the transition from the expanded to the compact conformation can induce a structural change in the

interprotomer binding site and, thus, a change in the dissociation constant of the Na⁺ binding. This promotes the release of the Na⁺ from the interprotomer binding site. Accordingly, a scenario that the "expanded"-to-"compact" switch regulates the Na⁺ transport channel through pIRC2 is possible. I suggest to include the above-mentioned discussion on the allosteric coupling in the manuscript.

Answers to Reviewers' comments:

Reviewer #3 (Remarks to the Author):

The authors have incorporated all changes requested by me. I only have two minor changes to add before the manuscript is ready for publication concerning the Computational part:

We thank the reviewer for his help in improving our manuscript. We reply to the comments and highlight all the changes in the text with yellow as follows:

-Methods: please give height and width of Gaussians that were placed in metadynamics runs.

The height and width of Gaussians were given in the Material and Methods (lines 503-504 of the originally submitted version) and were equal to 1kJ/mol and 0.05 nm, respectively.

-line 253: change "molecular dynamics" to "metadynamics". Furthermore include a sentence that you performed these simulations with the intent to find unbinding pathways, but not to determine free energies along them.

We thank the reviewer for this suggestion. We replaced "molecular dynamics" with "metadynamics" in the text. It is already indicated in the text, that we perform metadynamics simulations in order to probe the ion-release pathways, starting from the sodium-bound conformation (lines 253-254 of the original version of the manuscript). To avoid misunderstanding we included the following explanation into the Materials and Methods section: "Since the correct sampling of protein degrees of freedom relevant to the ion release at the late states of the photocycle appears problematic, we performed metadynamics simulations allowed us to reveal possible sodium unbinding pathways rather than quantitatively assess the free energies changes associated with them."

Reviewer #4 (Remarks to the Author):

In this paper, the authors present the structures of the O-state of the light-driven sodium ion-pumping rhodopsin KR2. The O-state is the key intermediate of Na⁺ pumping in KR2. The authors showed the protein structure for a physiologically relevant pentameric form. They identified the transient Na⁺ binding site inside the rhodopsin in the O-state and structural differences between the ground and the O-states. Together with the structures of the mutants, the authors discussed the protein structures to elucidate key determinants of cation pumping by KR2. The structural data provided in this paper are highly important and well-analyzed to understand the pathway of Na⁺ and the ion-pumping mechanism of KR2. Insights obtained by recent spectroscopic studies are closely related to the pathway and the ion-pumping mechanism proposed in the present study. Although the discussion in the manuscript is sound, the manuscript will be improved by the addition of the relating insights provided the spectroscopic studies. I suggest that I recommend the publication of this work in Nature Communications after major revisions indicated as follows.

We thank the reviewer for these valuable comments. Here is our reply to the comments. All the corresponding changes in the text are shown in yellow.

1. Line 145. The authors claimed that the retinal chromophore adopts a distorted all-trans configuration in the O-state. However, this is in contrast to the recently published time-resolved FTIR data on KR2. The discrepancy should be discussed more. Is the probability that the electron density is attributed to the 13-cis form completely ruled out? The authors mentioned the chromophore configuration in the O-state of GLR. But, this is a different protein. The discrepancy between the data reported for KR2 should be discussed more clearly.

Indeed it is an important question. Our electron density maps at 2.1 Å obtained with data corresponding to the 100% occupancy of the O-state clearly suggest all-trans configuration of the retinal chromophore in the O-state. However, as it was pointed out by the reviewer and also mentioned in our manuscript, this is in contrast to the recent time-resolved FTIR data on KR2. To validate the all-trans conformation we built omit and polder maps omitting retinal chromophore and K255 residue, which were shown in Fig. S2. The maps could be fitted satisfactorily by the all-trans, but not 13-cis configuration.

Nevertheless, to verify our conclusions once again and to address the question of the reviewer, we performed additional analysis of our crystallographic data and fitted the electron density maps with double all-trans/13-cis retinal with 100/0, 80/20, 60/40, 40/60, 20/80 and 0/100 ratios of all-trans/13-cis, respectively. The results are shown in Fig. 1. Fig. 1 is also incorporated into the Supplementary Fig. 2 of the manuscript as sections D and E. In brief, fitting of the data with more than 50% of the 13-cis retinal results in the negative difference electron densities near C_{14} and C_{20} atoms of the 13-cis retinal at the level higher than 3σ . Moreover, 13-cis configuration results in steric conflict between C_{15} and C_{20} atoms and nearby residues (W113 and W215, respectively).

Thus, our data strongly suggests that retinal is predominantly in the all-trans conformation in the O-state, although we cannot exclude the presence of a small fraction of the 13-cis retinal in the O-state of KR2.

Importantly, we should stress that the all-trans configurations of the retinal in the ground and O-states are different, which is demonstrated in Fig. S3. While in the ground state the planar organization of the molecule is slightly distorted, in the O-state retinal is almost completely planar, but is kinked notably near C_{14} atom. We suggest that this bending of the retinal may result in the appearance of the very broad peak at 940 cm^{-1} in the FTIR spectrum of the O-state of KR2 at 240 K, described in Chen et al., 2018. Unfortunately, we cannot also exclude either that the reason of the contradictions of our crystallographic data with the earlier obtained FTIR data may originate from the sample preparation and environment. Indeed, as a demonstration of the effect of different experimental conditions, we can mention that FTIR spectra of the H30A mutant, shown in Fig. 5d of the original work on KR2 (Inoue et al., 2013), indicate that H30A does not bind sodium in the ground state. Nevertheless, as it was demonstrated in our manuscript, crystal structure of the pentameric H30A reveals the sodium binding site the same to that of the wild type protein.

Following the suggestion of the reviewer we added this discussion of possible reasons for the discrepancy into the “Retinal binding pocket of KR2 in the O-state” section of the manuscript.

Fig. 1. Electron density maps around retinal cofactor. The maps were built using the data of the O-state of KR2 at 100K at 2.1 Å and 6 models with 100/0, 80/20, 60/40, 40/60, 20/80 and 0/100 proportions of all-*trans*/13-*cis* retinal configurations ratios. 2F_o-F_c maps are contoured at the level of 1.5 σ and are shown with gray mesh. Difference negative F_o-F_c maps are contoured at the level of 3 σ and are shown with red mesh. Bottom section shows the inadequately short distances between retinal C₁₅ and C₂₀ atoms to the nearby residues when fitting the data with 13-*cis* retinal.

2. Line 152. The authors claimed that relative locations of the RSBH⁺ and D116 side chain are similar to those in the ground state. This conclusion is quantitatively supported by the data of a recent resonance Raman study (Ref. #14). Ref. #14 reported that C=N stretching frequencies of the RSB are very similar between the ground state (1640 cm⁻¹) and the O-state (1642 cm⁻¹). The C=N stretching frequency is a sensitive marker for the hydrogen bond strength of the protonated RSB. The similar frequencies of the C=N stretching mode support the authors' claim that relative locations of the RSBH⁺ and D116 side chain are similar to

those in the ground state. I suggest to include the above-mentioned discussion on the C=N stretching frequencies in the manuscript.

We fully agree with the reviewer comment that the similar relative location of RSBH⁺ and D116 in the ground and the O-state is indeed supported in the Ref #14, which was mentioned in our manuscript. To make it more clear for the readers, we added the following sentences to the section “Retinal binding pocket of KR2 in the O-state”: “Indeed, in ¹¹ authors reported that C=N stretching frequencies of the RSB are very similar between the ground state (1640 cm⁻¹) and the O-state (1642 cm⁻¹). The C=N stretching frequency is a sensitive marker for the hydrogen bond strength of the protonated RSB. The similar frequencies of the C=N stretching mode support that relative locations of the RSBH⁺ and D116 side chain are similar to those in the ground state.”.

3. Line 184. The switching between the expanded and the compact conformations is interesting and reasonable to understand the ion pumping mechanism. A recent resonance Raman study (Otomo et al., Biochemistry 2020, 59, 520–529) showed that the Na⁺ binding to the interprotomer binding site changes the hydrogen bond strength of the protonated RSB in the ground state of KR2, suggesting an allosteric communication from the interprotomer binding site to the protonated RSB. If the interprotomer binding site and the RSB are allosterically coupled, the transition from the expanded to the compact conformation can induce a structural change in the interprotomer binding site and, thus, a change in the dissociation constant of the Na⁺ binding. This promotes the release of the Na⁺ from the interprotomer binding site. Accordingly, a scenario that the “expanded”-to-“compact” switch regulates the Na⁺ transport channel through pIRC2 is possible. I suggest to include the above-mentioned discussion on the allosteric coupling in the manuscript.

The work by Otomo et al. appeared very recently, so we could not mention it in the original version of the manuscript. We agree that this interesting work should be discussed in the manuscript.

We agree with the reviewer, that structural rearrangements in the RSB region upon Na⁺ release (and, at the same time ‘compact’-to-‘expanded’ conformational switch) may be allosterically coupled with the Na⁺ binding site at the KR2 surface. This, consequently, may lead to the unbinding of the ion from the site and its replacement by the ion, newly released from the KR2 protomer. In the revised version we have added the following discussion,

suggested by the Reviewer, to the “Sodium translocation pathway” section: “Recently, it was also shown that there is an allosteric communication between the interprotomer Na⁺ binding site and the RSB hydrogen bond already in the ground state²⁶. Thus, structural rearrangements of the RSB-counterion pair upon Na⁺ release and corresponding ‘compact’-to-‘expanded’ conformational switch may allosterically affect the interprotomer Na⁺ binding site, promoting Na⁺ unbinding from the site, observed in metadynamics simulations.”.

We also briefly mentioned the recent study (Otomo et al.) in the “Summary” section.

REVIEWERS' COMMENTS:

Reviewer #4 (Remarks to the Author):

The manuscript has been revised well. I think this manuscript is acceptable as it is.